# Changes in the distribution of the tear film lipid layer after intensive pulsed light combined with meibomian gland expression in patients with meibomian gland dysfunction

**Yongwoo Lee[1,2], Sung Woo Lee[3,4], Ji Kyu Yun[1], So Young Han[5], Chul Young Choi[5]***

**1** Department of Ophthalmology, Kangwon National University Hospital, Kangwon National University School of Medicine, Chuncheon, Republic of Korea, **2** Department of Ophthalmology, Columbia University Irving Medical Center, Vagelos College of Physicians and Surgeons, Columbia University, New York, New York, United States of America, **3** Global Leadership College, Yonsei University, Seoul, Republic of Korea, **4** Snow Subsidiary Company, Naver Corporation, Seongnam, Republic of Korea, **5** Department of Ophthalmology, Kangbuk Samsung Hospital, Sungkyunkwan University School of Medicine, Seoul, Republic of Korea

* sashimi0@naver.com

## Abstract

### Purpose

This study aimed to investigate changes in dry eye disease (DED) parameters and tear film lipid layer distribution after intensive pulse light (IPL) combined with meibomian gland expression (MGX) in patients with meibomian gland dysfunction (MGD).

### Methods

This retrospective study included 218 patients diagnosed with MGD who underwent IPL combined with MGX. Various DED parameters, including tear film lipid layer thickness (LLT), were measured using a Placido disc tear film analyzer and slit lamp. Inferior corneal images were quantified and divided into upper, lower, nasal, and temporal quadrants, with further subdivision into six parts from top to bottom using Python.

### Results

The ocular surface disease index, meibomian gland expressibility, and quality scores significantly improved after three treatment sessions. Slit-lamp-measured DED parameters also improved, excluding the fluorescein-stained tear meniscus height. Redness in the nasal limbal and bulbar conjunctivae significantly decreased. The mean LLT tended to increase after treatment. LLT in the upper half of the images, but not in the lower half, increased significantly, with the difference in LLT between the two halves decreasing significantly from 34.46 ± 15.73 to 30.27 ± 14.63 nm (p = 0.031). When the vertical distribution was analyzed by further subdivision into six equal parts from top to bottom, the average difference in LLT decreased in the uppermost segment after treatment.

**Data availability statement:** Research data may contain personal information and cannot be disclosed for ethical reasons. These restrictions have been imposed by the Kangwon National University Hospital Institutional Review Board to protect potentially identifying or sensitive patient information. However, if there is a reasonable request, it can be requested through the Kangwon National University Hospital Institutional Review Board. (knuh.irb@gmail.com)

**Funding:** This study was supported by Kangwon National University Hospital in 2023 (to author YL).

**Competing interests:** The authors have declared that no competing interests exist.

## Conclusion

IPL combined with MGX reduced the vertical distribution of lipids in patients with MGD by reducing lower tear film lipid layer stasis. Thus, the Placido disc tear film analyzer is a useful tool for analyzing lipid layer distribution in MGD.

## Introduction

Dry eye disease (DED) is an intractable multifactorial disease of the tear film and ocular surface. While DED is the most common reason for patients seeking ophthalmological care, the mechanism and target for treatment remain unclear [1]. In 2017, the Tear Film and Ocular Surface Society presented a contemporary definition and classification system for DED and defined evaporative DED on the basis of meibomian gland dysfunction (MGD). Since then, the tear film lipid layer has emerged as a target for DED treatment [2].

Currently, intensive pulse light (IPL) combined with meibomian gland expression (MGX) is widely used as a treatment for MGD, and there are many reports of improvements in dry eye symptoms and signs following the use of this treatment [3–5]. However, evaluating the effects of IPL therapy using a single parameter is difficult because the symptoms and signs are often inconsistent [6]. Although an interferometer, such as Lipiview® by TearScience Inc., can be used to quantitatively analyze the tear film lipid layer thickness (LLT), it cannot determine improvements in MGD merely by analyzing LLT [7].

Recently, a Placido disc tear film analyzer was used to objectively measure the noninvasive tear break-up time (NIBUT), noninvasive tear meniscus height (NITMH), and conjunctival redness. By generating the interference image of the tear lipid layer, it was possible to measure LLT by analyzing the color of the image [8].

As a change in the average tear film LLT is an important marker of MGD-induced DED, we hypothesized that the distribution of the lipid layer is equally important [8]. However, to the best of our knowledge, no studies have investigated the tear film lipid layer distribution following treatment with IPL and MGX.

Therefore, the aim of this study was to determine changes in the distribution of the tear film lipid layer by dividing and analyzing lipid layer images along with various DED parameters using a Placido disc tear film analyzer before and after IPL combined with MGX in patients with MGD.

## Materials and methods

### Participants

Fig 1 summarizes the overall study protocol in a flowchart. This study was approved by the Kangwon National University Institutional Review Board (No. KNUH-2023-06-006) and conducted in accordance with the principles of the Declaration of Helsinki. Consent was waived by Kangwon National University Institutional Review Board due to the nature of a retrospective study. All data were fully anonymized before access and analysis, ensuring that no personal identifiers were available. We retrospectively analyzed data for all patients with MGD who visited the Department of Ophthalmology and received IPL combined with MGX from November 1, 2022 to May 31, 2023 at Kangwon National University Hospital, South Korea. Their clinical data were accessed on July 22, 2023. MGD diagnosis, evaluation, and classification were based on criteria outlined at the International Workshop on MGD conducted by the Tear Film and Ocular Surface Society [9]. The medical

## Study subjects
- Patients diagnosed with meibomian gland dysfunction (MGD) who visited the hospital for treatment.
- No history of corneal surgery or other ocular disease
- 20≤ Age ≤75 years

↓

## Screening tests (Baseline data)
- Ocular Surface Disease Index (OSDI)
- Placido tear film analyzer: Noninvasive break-up time (NIBUT), noninvasive tear meniscus height (NITMH), redness score, lipid layer thickness (LLT), and meibography.
- Fluorescein break-up time (FBUT), fluorescein tear meniscus height (FTMH), stain score, and MGD evaluation.

↓

## Intensive pulse light combined with meibomian gland expression (3 times at 2-week intervals)

↓

## Outcome tests (2-weeks after the last treatment)
- OSDI
- Placido tear film analyzer: NIBUT, NITMH, Redness score, LLT, Meibography
- FBUT, FTMH, stain score, and MGD evaluation

↓

## Data organization and analysis
- Pre- and post-treatment effects analyzed using paired t-tests.

**Fig 1. Flowchart of overall study protocol.**

charts of a total of 337 patients were reviewed, and the following criteria were used for exclusion: (1) age < 20 or > 75 years; (2) history of refractive corneal surgery; (3) other ophthalmic diseases, such as glaucoma or retinal disease; (4) administration of glaucoma eye drops, antibiotics, or anti-inflammatory drugs within 3 months; and (5) missed clinical examinations before and after IPL treatment. If one eye met these criteria, both eyes were excluded.

## Clinical examination

All patients who received IPL combined with MGX underwent a basic examination after a washout period of 4 weeks. The patient's history and Ocular Surface Disease Index (OSDI) questionnaire were used to assess DED symptoms. Before the eye-touching test was performed, NIBUT, NITMH, redness, and LLT were measured by an experienced ophthalmology technician using a Placido disc tear film analyzer (Keratograph 5M®, Oculus, Wetzlar, Germany).

The fluorescein-stained tear break-up time (FBUT), fluorescein-stained tear meniscus height (FTMH), corneal and conjunctival fluorescein scores according to the Oxford scale, and MGD were evaluated under a slit-lamp microscope. Meibography images were obtained to calculate the ratio of the area of the meibomian glands to the total area using ImageJ software (National Institutes of Health, Bethesda, MD, USA).

IPL combined with MGX was performed three times at 2-week intervals. Two weeks after the final treatment, OSDI scores, Placido tear film analyzer findings, DED parameters, and MGD signs were measured using the same process.

## Treatment protocol

An experienced surgeon (YL) performed IPL combined with MGX three times at 2-week intervals, in accordance with the protocol described in a previous study by Toyos et al [10]. However, because excessive meibum expression immediately after treatment can compromise the reliability of examination results, evaluations were conducted 2 weeks after the final treatment session. Fitzpatrick skin typing was performed before the first treatment. IPL (M22, Lumenis Ltd., Yokneam, Israel) was applied with appropriate energy using an acne filter (wavelengths of 400–600 and 800–1,200 nm) [3] with a 15- × 35-mm tip. The power was adjusted to 10–12 J/cm² according to the skin type and patient response, with 10–12 pulses administered per pass. After lubricant gel application, two passes were administered to the upper and lower eyelids. When administering IPL to the upper eyelid, a metal spatula was gently inserted between the upper eyelid and bulbar conjunctiva to protect the eyeball before the first pass. After the second IPL pass on both eyes, MGX was performed by gentle compression of the tarsal conjunctivae of the upper and lower eyelids using a cotton swab.

## Lipid layer image analysis

Tear film lipid layer images were quantitatively analyzed and segmented by a program specialist as previously described [8]. LLT was estimated by referencing the color of each pixel (red, green, and blue scales) to a look-up table and approximating it using the principle of the nearest Euclidean distance [11] as follows.

$$d = \sqrt{(r-R)^2 + (g-G)^2 + (b-B)^2}$$

where the lowercase letters (r, g, b) represent the points on the inspected image and uppercase letters (R, G, B) represent the points on the look-up table.

All images were extracted 2 s after two spontaneous blinks, and LLT was measured using the Placido disc tear film analyzer (Keratograph 5M®, Oculus) in the lower half of the cornea. All images were uniformly cropped using the ninth Placido ring as an analysis reference [8].The upper cornea was excluded because of the possibility of irregular measurements due to the influence of the eyelids or eyelashes. The total LLT was calculated before image segmentation. For analysis of the regional distribution of lipid layer, the extracted images were segmented using Python (Pillow Library, version 8.4.0, Python Software Foundation,

Wilmington, DE, USA). After image height measurement, the image was divided into upper and lower halves using an image cropping tool, and batch cropping was performed. Differences in LLT between the upper and lower halves were calculated. The width of the image was then analyzed in a similar manner after dividing the lipid layer into halves on the nasal and temporal sides using the cropping tool, with further subdivision into quarters. To subdivide and measure the vertical distribution, the change in lipid layer distribution was evaluated in six sections created from top to bottom for analysis.

## Statistical analysis

Using G*Power software (version 3.1.9.7, Heinrich-Heine-Universität Düsseldorf, Germany), we determined that a minimum sample size of 34 participants was required to achieve 80% statistical power with a significance level of 0.05 and an estimated effect size of 0.5. The paired t-test was used to compare all test results obtained before and after treatment. Statistical analyses were performed using SPSS software (version 26.0; IBM Corp., Armonk, NY, USA), and p < 0.05 was considered statistically significant. Among the two eyes of each patient, only the eye with a higher MGD grade was included in the statistical analysis.

## Results

Overall, 337 patients were enrolled, of which, 218 patients (218 eyes; 136 women; mean age, 61.31 ± 15.04 [range, 34–83] years) met the inclusion criteria (Table 1).

Compared with those before treatment, OSDI scores decreased significantly after treatment, from 41.90 ± 23.19 to 36.58 ± 23.88 (p < 0.001). Moreover, meibomian gland expressibility and quality scores significantly decreased after treatment (p < 0.001). FBUT was significantly increased (p < 0.001), whereas no significant differences (p = 0.598) were observed in FTMH. The keratoconjunctival staining score also showed a significant decrease after treatment (Table 2). In the dry eye index test using the Placido disc tear film analyzer, the mean NITMH and NIBUT increased slightly, however, the difference was not significant. While, no significant differences were found in total and temporal conjunctival redness, nasal bulbar conjunctival redness and nasal limbal conjunctival redness decreased significantly from 1.90 ± 0.78 to 1.73 ± 0.72 (p = 0.003) and from 1.43 ± 0.68 to 1.24 ± 0.64 (p = 0.001), respectively. The percentage of the meibomian gland area on the meibography images was not significantly different before and after treatment (p = 0.852; Table 2). In the lipid layer examination, the total LLT slightly increased from 65.26 ± 14.33 nm before treatment to 66.80 ± 11.68 nm after treatment; however, this increase was not significant (p = 0.168). When the upper, lower, nasal, and temporal quadrants were assessed, LLT in the two upper regions (superotemporal and superonasal) increased significantly after treatment (p = 0.003 and 0.022, respectively), whereas the differences in the two lower regions (inferotemporal and inferonasal) was not significant (p = 0.203 and 0.302, respectively). The difference in the

**Table 1. Patient demographics.**

| Characteristics | Values |
| --- | --- |
| No. of patients/eyes | 218 |
| Age (years) | 61.31 ± 15.04 (34–83)* |
| Sex (male) | 37.6% (82/218) |
| Fitzpatrick score | 3.55 ± 0.33 (1–5) |

Mean ± standard deviation (MIN–MAX).

thicknesses of the upper and lower lipid layers significantly decreased from 19.65 ± 8.85 to 17.56 ± 8.47 nm after treatment (p = 0.031; Table 2).

In addition, LLT was thicker in the lower part than in the upper part and on the temporal side than on the nasal side (Fig 2). This tendency became more evident when the vertical distribution was analyzed by further subdivision into six equal parts from top to bottom. After treatment, the average difference in LLT decreased in the uppermost segment (Fig 3).

## Discussion

Recently, our understanding of MGD has increased. Many favorable therapeutic effects of IPL combined with MGX have been reported, and this treatment has been established as a representative treatment for MGD-related DED [12].

**Table 2. Changes in symptom scores and signs of dry eye disease and meibomian gland dysfunction after treatment with intensive pulse light and meibomian gland expression (n = 218).**

| | Pre IPL + MGX | Post IPL + MGX | p-value* |
|---|---|---|---|
| OSDI | 41.90 ± 23.19 | 36.58 ± 23.88 | **<0.001** |
| MGD grade | | | |
| Expressibility | 1.81 ± 0.81 | 1.31 ± 1.02 | **<0.001** |
| Quality | 2.19 ± 0.53 | 1.48 ± 0.87 | **<0.001** |
| Telangiectasis | 1.71 ± 0.45 | 1.54 ± 0.50 | **0.004** |
| FBUT (s) | 4.38 ± 1.07 | 5.85 ± 2.18 | **<0.001** |
| FTMH (mm) | 0.18 ± 0.04 | 0.19 ± 0.03 | 0.598 |
| Stain score | | | |
| Cornea | 1.41 ± 0.80 | 0.92 ± 0.50 | **0.022** |
| Conjunctiva | 0.75 ± 0.72 | 0.42 ± 0.49 | **<0.001** |
| Tear film analyzer | | | |
| NITMH (mm) | 0.24 ± 0.12 | 0.26 ± 0.08 | 0.097 |
| NIBUT first (s) | 6.17 ± 3.89 | 6.74 ± 4.30 | 0.125 |
| NIBUT average (s) | 10.16 ± 5.10 | 10.36 ± 4.95 | 0.649 |
| Redness | | | |
| Total | 1.74 ± 0.57 | 1.67 ± 0.59 | 0.063 |
| Bulbar temporal | 1.67 ± 0.55 | 1.67 ± 0.64 | 0.936 |
| Bulbar nasal | 1.90 ± 0.78 | 1.73 ± 0.72 | **0.003** |
| Limbal temporal | 1.18 ± 0.43 | 1.18 ± 0.57 | 0.840 |
| Limbal nasal | 1.43 ± 0.68 | 1.24 ± 0.64 | **0.001** |
| Meibography (%) | 59.10 ± 11.08 | 59.28 ± 10.91 | 0.852 |
| LLT (nm) | | | |
| Total | 65.26 ± 14.33 | 66.80 ± 11.68 | 0.168 |
| LLT inferotemporal | 80.41 ± 15.12 | 81.81 ± 15.80 | 0.203 |
| LLT superotemporal | 57.88 ± 14.92 | 61.65 ± 12.94 | **0.003** |
| LLT inferonasal | 72.04 ± 17.88 | 72.51 ± 15.82 | 0.302 |
| LLT superonasal | 55.50 ± 15.96 | 58.46 ± 12.89 | **0.022** |
| Vertical difference | 19.65 ± 8.85 | 17.56 ± 8.47 | **0.031** |
| Horizontal difference | 7.73 ± 6.35 | 8.58 ± 6.29 | 0.170 |

* Paired t-test; bold indicates statistical significance. Abbreviations: IPL, intensive pulse light; MGX, meibomian gland expression; OSDI, ocular surface disease index; MGD, meibomian gland dysfunction; FBUT, fluorescein-stained tear break-up time; FTMH, fluorescein-stained tear meniscus height; NITMH, non-invasive tear meniscus height; NIBUT, non-invasive tear break-up time; LLT, lipid-layer thickness.

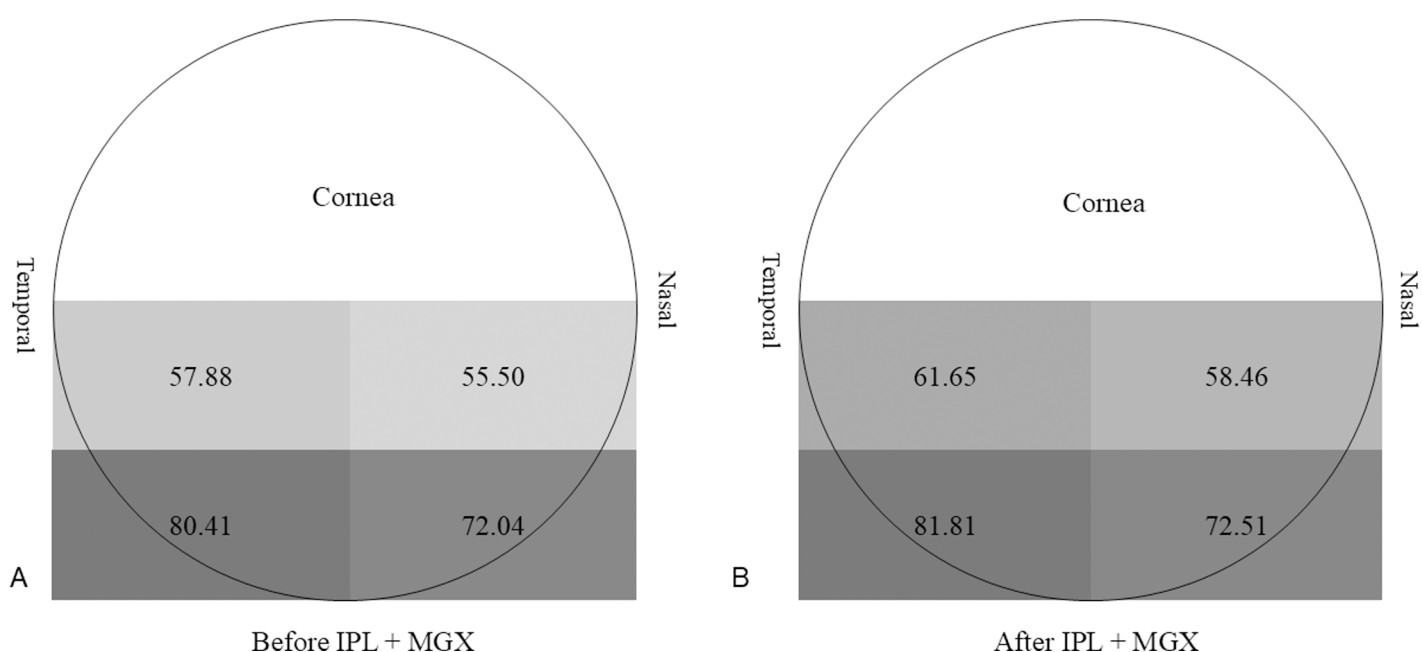

**Fig 2. Lipid layer thickness distribution in the inferior half of cornea (nm).** Lipid layer thickness distribution in the inferior cornea (nm) was measured using a Placido disc tear film analyzer before and after intensive pulse light combined with meibomian gland expression. The color of the shading reflects the thickness of the lipid layer.

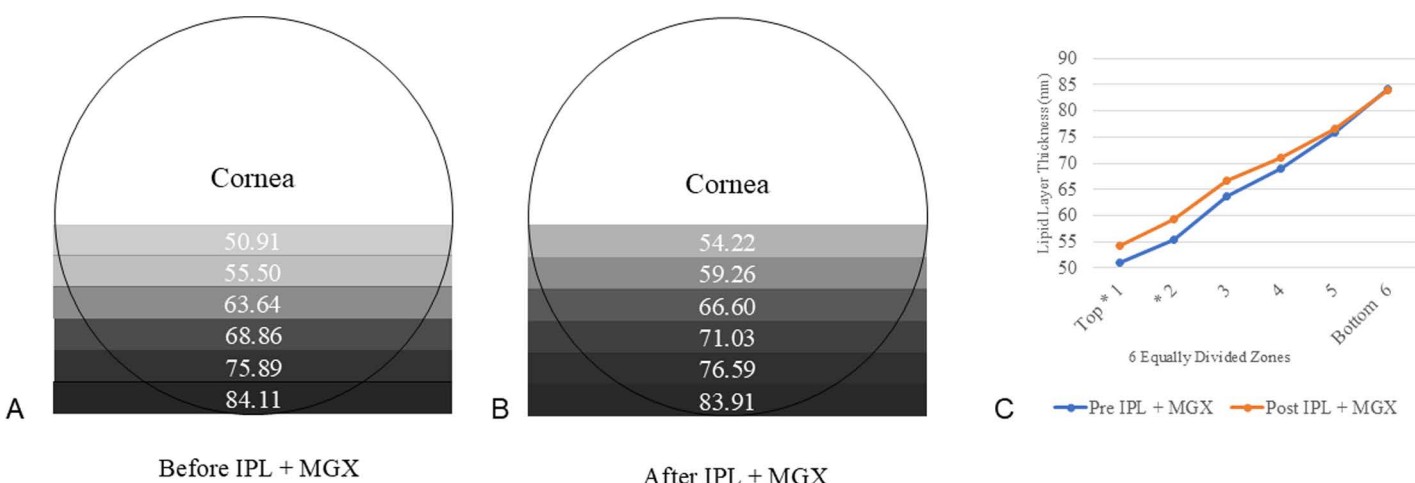

**Fig 3. Change in the vertical distribution of the lipid layer thickness after intensive pulse light treatment (IPL; nm) combined with meibomian gland expression (MGX).** Vertical distribution of the tear film lipid layer was measured before and after treatment. The color of the shading reflects the thickness of the lipid layer and indicates a decrease from the upper to the lower region after treatment. * Statistically significant difference ($p < 0.05/6$ Bonferroni correction, paired t-test).

LLT is an important index for evaluating treatment response and improvements in MGD, and several treatments for MGD were documented to improve LLT [13,14]. However, evaluating MGD severity using only LLT is insufficient, as MGD-related DED can occur even in patients with a normal LLT [15–17]. While LLT is a parameter for evaluating treatment response in DED [7], it does not always reflect symptom severity, as demonstrated in this study, where the mean LLT increased slightly without significance after IPL

combined with MGX (Table 2). This highlights the need for the qualitative analysis of lipid layer distribution, which has not been previously examined after treatment with IPL and MGX.

A Placido disc tear film analyzer does not quantitatively determine the lipid layer; rather, it allows its visualization as an image. However, we quantitatively analyzed the lipid layer in the area shown on the image using a lipid layer analysis program [8]. Specifically, we measured the average LLT using a lipid layer image and analyzed it in detail.

To compare the lipid layer distribution, we captured images and divided the lipid layer into upper, lower, nasal, and temporal quadrants. Although the difference in LLT between the nasal and temporal quadrants was not significant, the difference between the upper and lower quadrant LLT decreased after treatment with IPL and MGX (Table 2).

Thus, we hypothesized that LLT may show more even vertical distribution after IPL combined with MGX. After further analysis of the lipid layer images by subdividing them into six segments from top to bottom, the improvements in the lipid layer was more evident toward the upper segment than the lower segment. This occurs because blinking spreads the lipid layer evenly across the cornea. Overall, we found that lipid layer distribution is an important factor for improving the MGD [18], particularly in the upper part of the image, because MGD was alleviated after treatment and the stagnation phenomenon of the lower lipid layer was reduced. In addition, the improvement observed in the lipid layer and visual function following combined IPL and MGX may be attributed to tear film stabilization in the area adjacent to the pupil [19]. To fully validate this hypothesis, the lipid layer across the entire cornea should be analyzed. However, in the present study, the upper cornea was excluded to ensure measurement reliability. Future studies are warranted to comprehensively evaluate the lipid layer distribution, including that in the upper cornea.

Furthermore, changes in the composition of meibum secretions may have resulted in the spread of the top lipid layers. The tear film is composed of an outermost lipid layer, which stabilizes the tear film by reducing the surface tension and suppressing evaporation of the inner aqueous layer. Phospholipids such as phosphatidylcholine, palmitic-acid-9-hydroxy-stearic-acid, and O-acyl-ω-hydroxy fatty acids have recently been identified as polar lipid layers of the tear film and account for approximately 20 mol% of the inner layer [20–22]. Wax esters and cholesteryl esters are representative of the nonpolar lipid layer and account for 80 mol% of the outer layer [21]. However, the effects of pathological changes in the tear film lipid layer components on surface tension and tear film rheology and MGD development remain unknown.

Studies have reported altered sphingolipid composition of the meibum in patients with MGD [23], as well as a correlation between free fatty acid composition and color in human meibum [24]. In the present study, MGD was alleviated after treatment with IPL and MGX and tear film lipid layer distribution increased evenly toward the upper part. These findings, rather than LLT alone, are important for understanding MGD pathophysiology and treatment response. It can be inferred that this treatment caused a change in the composition of meibum to reduce surface tension and alleviate MGD.

One notable finding in this study is that only nasal conjunctival redness showed a significant decrease after treatment. While previous studies have reported that nasal conjunctival hyperemia may exhibit different patterns even in healthy individuals [25], the present study raises the possibility that conjunctival hyperemia may vary between the nasal and temporal regions in patients with dry eye. Further research is warranted to explore the underlying causes of this observation.

We found significant improvements in the distribution of the tear film lipid layer following IPL combined with MGX, particularly in the vertical distribution. Song et al. [26] emphasized

the potential of IPL therapy to enhance tear film stability by altering the lipid layer profile. Similarly, Ahmed et al. [27] demonstrated changes in tear proteins and lipids after IPL therapy, suggesting a mechanistic link between IPL therapy and improved tear film composition. Zhao et al. further revealed alterations in lipid profiles, elucidating the biochemical basis of tear film stabilization following IPL treatment [28]. In the present study, while the change in the overall LLT was not significant, the decrease in the difference between upper and lower LLT after treatment suggests improved lipid layer distribution, which may reflect reduced stagnation and better spread of tear lipids. These findings contribute to the growing body of evidence supporting the role of IPL treatment in enhancing tear film quality through lipid layer modulation.

Further investigation into the relationship between LLT changes and clinical parameters of DED is warranted. For instance, exploring the correlations between LLT uniformity and subjective symptom improvement, such as OSDI score reductions, may provide deeper insights into the clinical relevance of IPL-induced lipid layer changes. In addition, the evaluation of horizontal LLT differences, as suggested by Zhao et al., could help identify patterns of improvement that were not captured in this study.

In this study, LLT was smaller on the nasal side than on the temporal side in the basic analysis of lipid layer distribution (Fig 2). We can infer that LLT improvements on the nasal side (where LLT is relatively smaller because of MGD alleviation following IPL treatment) influences the alleviation of nasal conjunctival injection; however, further studies are needed.

This study has some limitations. The observation period was short, and the components of tear film lipids were not analyzed. Further research over a longer period of time should investigate these aspects. Furthermore, this study did not include a control group without IPL combined with MGX; therefore, future prospective comparative studies are warranted to validate these findings. Additionally, it would be valuable to compare the tear film LLT measured in this study with data obtained using widely utilized interferometers in future research.

To the best of our knowledge, this is the first study to analyze the lipid layer by division into upper, lower, nasal, and temporal quadrants. In the future, we plan to conduct a prospective study with patients to determine the validity of our results.

## Conclusions

As MGD was alleviated after treatment with IPL and MGX, an increase in LLT, which was more pronounced in the upper portion of the inferior half of the cornea, was observed. It can be inferred that the distribution of the lipid layer improved more evenly after MGD treatment. Our findings also indicate that the Placido tear film analyzer is a useful instrument for segmenting and analyzing tear film lipid layers. In the future, qualitative analysis of LLT and tear film lipid layer distribution will play an important role in the treatment of MGD-related DED.

## Author contributions

**Conceptualization:** Yongwoo Lee.

**Data curation:** Yongwoo Lee, Ji Kyu Yun.

**Formal analysis:** Sung Woo Lee.

**Resources:** Yongwoo Lee.

**Software:** Sung Woo Lee.

**Writing – original draft:** Yongwoo Lee, Chul Young Choi.

**Writing – review & editing:** Yongwoo Lee, So Young Han, Chul Young Choi.

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
