## [Decision Letter · Decision Letter 0]

10 Jan 2025

PONE-D-24-24404

Changes in the distribution of the tear lipid layer after intensive pulse light combined meibomian gland expression in patients with meibomian gland dysfunction

PLOS ONE

Dear Dr. Choi,

Thank you for submitting your manuscript to PLOS ONE. After careful consideration, we feel that it has merit but does not fully meet PLOS ONE’s publication criteria as it currently stands. Therefore, we invite you to submit a revised version of the manuscript that addresses the points raised during the review process.

We look forward to receiving your revised manuscript.

Kind regards,

Helen Howard

Staff Editor

PLOS ONE

Journal Requirements:

For additional information about PLOS ONE ethical requirements for human subjects research, please refer to http://journals.plos.org/plosone/s/submission-guidelines#loc-human-subjects-research .

3. For studies involving third-party data, we encourage authors to share any data specific to their analyses that they can legally distribute. PLOS recognizes, however, that authors may be using third-party data they do not have the rights to share. When third-party data cannot be publicly shared, authors must provide all information necessary for interested researchers to apply to gain access to the data. (https://journals.plos.org/plosone/s/data-availability#loc-acceptable-data-access-restrictions)

Additional Editor Comments:

The manuscript has been evaluated by three reviewers, and their comments are available below.

The reviewers have raised a number of concerns that need attention. They request additional information on methodological aspects of the study, revisions to the statistical analyses and improved discussion.

Could you please revise the manuscript to carefully address the concerns raised?

Comments from PLOS Editorial Office: We note that one or more reviewers has recommended that you cite specific previously published works. As always, we recommend that you please review and evaluate the requested works to determine whether they are relevant and should be cited. It is not a requirement to cite these works. We appreciate your attention to this request.

Reviewers' comments:

Reviewer's Responses to Questions

**Comments to the Author**

1. Is the manuscript technically sound, and do the data support the conclusions?

Reviewer #1: No

Reviewer #2: Yes

Reviewer #3: No

2. Has the statistical analysis been performed appropriately and rigorously? 

Reviewer #1: Yes

Reviewer #2: I Don't Know

Reviewer #3: I Don't Know

3. Have the authors made all data underlying the findings in their manuscript fully available?

Reviewer #1: Yes

Reviewer #2: Yes

Reviewer #3: No

4. Is the manuscript presented in an intelligible fashion and written in standard English?

Reviewer #1: Yes

Reviewer #2: Yes

Reviewer #3: No

5. Review Comments to the Author

Reviewer #1: The purpose of this retrospective study was to examine changes in DED characteristics and lipid-layer distribution after IPL and MGX treatment in MGD patients. They examined 218 MGD patients who received IPL and MGX therapy. A Placido-disc-based tear film analyzer and slit lamp were used to assess differences in DED characteristics, LLT, avg LLT differences in inferior corneal pictures were evaluated by splitting them into upper, lower, nasal, and temporal quadrants, then into six sections from top to bottom using Python. After 3 IPL therapy OSDI, MG expressibility, and quality scores improved. DED symptoms improved, except for fluorescein-stained tear meniscus height. Redness in the nasal limbal and bulbar conjunctiva decreased markedly, but not in the total conjunctiva. IPL therapy increased mean LLT. The LLT grew dramatically in the upper half of the photos, but not in the lower half. The difference in LLT between halves reduced significantly. Post-IPL treatment, lipid-layer distribution discrepancies diminished in the uppermost region after picture segmentation. The study concludes that IPL and MGX treatment improved lower tear lipid-layer stasis and vertical lipid distribution in MGD patients. The Placido tear film analyzer can be used for assessing lipid-layer distribution in MGD.

I would like to compliment the authors on their interesting manuscript. However, several points and comments have to taken into consideration prior to publication.

1. Lines 31-32: “Slit-lamp-measured DED signs also increased, except for the fluorescein-stained tear meniscus height.” Please clarify, rather than increase/decrease, please write improved/worsened for clarity.

2. Lines 24: The study lacks a control group for definitive clinical conclusions. Please justify the lack of a control group and conclusions of the study.

3. Lines 106: Please clarify this “Lipid-layer image analysis”, aspect of the study. It’s not clear whether the authors are assessing the colors changes or deriving the thickness of TFLL from the interferometric patters obtained by the Keratograph 5M. While its clear that color changes equate to LLT please explain for readers how the changes equate to thickenss.

4. Furthermore, is the Total LLT= LLT inferotemporal+LLT superotemporal + LLT inferonasal + LLT superonasal or the mean of all these regions?

5. Please justify the use of lower half of the cornea for image analysis and its clincal significance.

6. It would be more substantial if the authors could compare their LLT with devices such as Lipiview, etc. If not possible please discuss.

7. Table 2 shows the changes in various clinical characteristics of dry eye, it hasn’t been mentioned how these changes effects the LLT regions or Horizontal difference of LLT. Possible this might bring out the clinical significance of this study, any correlation or risk factors. Please include it in the discussion:

a. Song, Yilin, et al. "Tear film interferometry assessment after intense pulsed light in dry eye disease: A randomized, single masked, sham-controlled study." Contact Lens and Anterior Eye 45.4 (2022): 101499.

b. Ahmed, S. et al. (2019). Effect of intense pulsed light therapy on tear proteins and lipids in meibomian gland dysfunction. Journal of ophthalmic & vision research, 14(1), 3.

c. Zhao, Hui, et al. "Lipidomics profiles revealed alterations in patients with meibomian gland dysfunction after exposure to intense pulsed light." Frontiers in Neurology 13 (2022): 827544.

8. Figure 1 should a detailed “study design and process” figure.

9. While figure 1 and figure 2 are interesting, these are averaging the pixels, please state clearly how many pixels were used to make each quadrant. How did the authors eliminate biases while performing this analysis as not all eyes are of the same size and the focal length of the patient is not regular. Furthermore, the iris of patients are of various color intensities.

10. Figure 2 contains 3 panels, please simplify by labeling them as A, B, C. Same with figure 1 that has 2 panels.

11. Figure 2 C is a line graph that has to has to have more description. i.e the x-axis and y-axis are not apparently clear to the reader.

12. Lines: 204. The hypothesis has not taken into consideration that the upper half of the cornea was not assessed and what are the pro and cons of this approach.

13. As stated in lines 224 this is a novel study, however, the lipid layer is a structure floating on the aqueous and mucin layers, the author has to assess the interplay of the various layers and whether the other layers (TMH volume, TBUT integrity, MG expressibility, MGquality had any influence on their findings.

Reviewer #2: I sincerely appreciate the opportunity to read and evaluate the authors research on tear lipid layer with IPL and meibomian gland expression. I’ve been privileged to review several dozen dry eye papers to date in addition to publishing several on the topic; in light of this, and although there may be hetereogeneity in measurements, patients, and follow up, this paper ranks among the top 1% on the topic in terms of scientific rigor and methodology. Very impressive, and a congratulations to the authors are in order.

Please see below for my comments:

Abstract: Line 50 should read“is a widely used treatment…” remove the word “most”

Intro does a great job at laying out the landscape of dry eye and adequately addresses the niche and need for this paper.

Methodology includes IRB and ethics review.

Dry eye protocols are quite rigorous and standardized, very impressive.

Language is concise.

Results: While it does say who did the IPL, does it say who did the exams?

The sole critique and question that I have is maybe the authors should include the NNT statistic based on the primary outcome. Also for those statistically significant differences on secondary outcomes was Bonferroni performed? If not, you may see that statistical significance disappeares due to the small difference between the groups, and the fact that it helps to adjust p-values and mitigate the risk of Type I errors arising from conducting numerous statistical tests on the same dataset.

Otherwise a brilliant study and look forward to the response.

Reviewer #3: Friday, January 3

Dear Editor

Thank you for the invitation to review the manuscript titled: Changes in the distribution of the tear lipid layer after intensive pulse light combined meibomian gland expression in patients with meibomian gland dysfunction.

The aim of the study is to investigate changes in dry eye disease (DED) parameters and lipid-layer distribution after treatment with intensive pulse light combined with Meibomian gland expression treatment in patients with Meibomian gland dysfunction (MGD). After reviewing the manuscript, I think it requires a major revision to be suitable for publication in your admired journal. However, some comments to the author are presented below to improve the manuscript:

Materials and Methods

1. Page 5, lines 39-35: please clear the treatment periods (What is the interval between sessions, and why were the measurements done two weeks after IPL, not earlier?

2. The method should explain the light parameters used during IPL therapy using an acne filter, such as the emitted light wavelength, light intensity, operating voltage, power, frequency, maximum optical energy, pulse duration, and repetition time or any data regarding the instrument used.

3. The author mentioned in the abstract that: This retrospective study included 218 patients diagnosed with MGD, while in the result section (Page 7, line 128), he mentioned the number of patients was 337. In addition, the number of patients should be added obviously in the method section.

4. In the discussion section, pages 10-11, from lines 182-194, there are obvious overlaps. These paragraphs can be summarized in one or two sentences only.

5. Page 13, lines 232-236: there was no relation between the two sentences about the treated eyes and the healthy individuals. Please try to find another reference to support your data.

6. Page 13, lines 241-242. The distribution of the lipids layer is affected by blinking as each time the eye blinks, the lipid layer on the surface of the eye is compressed and then steadily expands, which creates a non-equilibrium state, and affects the reorganization of the lipid layers. The author should enrich the discussion with previous studies regarding tear lipids as PMID: 30820280 and other previous studies.

7. The results obtained are not discussed carefully in the discussion section in addition to the weaknesses in the correlation between the obtained results and the previous literature.

Best Regards

6. PLOS authors have the option to publish the peer review history of their article (what does this mean? ). If published, this will include your full peer review and any attached files.

**Do you want your identity to be public for this peer review?** For information about this choice, including consent withdrawal, please see our Privacy Policy .

Reviewer #1: No

Reviewer #2: No

Reviewer #3: No

---

## [Author Response · Author response to Decision Letter 1]

20 Jan 2025

Reviewer #1: The purpose of this retrospective study was to examine changes in DED characteristics and lipid-layer distribution after IPL and MGX treatment in MGD patients. They examined 218 MGD patients who received IPL and MGX therapy. A Placido-disc-based tear film analyzer and slit lamp were used to assess differences in DED characteristics, LLT, avg LLT differences in inferior corneal pictures were evaluated by splitting them into upper, lower, nasal, and temporal quadrants, then into six sections from top to bottom using Python. After 3 IPL therapy OSDI, MG expressibility, and quality scores improved. DED symptoms improved, except for fluorescein-stained tear meniscus height. Redness in the nasal limbal and bulbar conjunctiva decreased markedly, but not in the total conjunctiva. IPL therapy increased mean LLT. The LLT grew dramatically in the upper half of the photos, but not in the lower half. The difference in LLT between halves reduced significantly. Post-IPL treatment, lipid-layer distribution discrepancies diminished in the uppermost region after picture segmentation. The study concludes that IPL and MGX treatment improved lower tear lipid-layer stasis and vertical lipid distribution in MGD patients. The Placido tear film analyzer can be used for assessing lipid-layer distribution in MGD.

I would like to compliment the authors on their interesting manuscript. However, several points and comments have to taken into consideration prior to publication.

 Thank you for providing valuable feedback and for your detailed review of our research. We will do our utmost to make improvements to enhance the quality of our study.

1. Lines 31-32: “Slit-lamp-measured DED signs also increased, except for the fluorescein-stained tear meniscus height.” Please clarify, rather than increase/decrease, please write improved/worsened for clarity.

 We have revised the terms "increased/decreased" to "improved/worsened" in the manuscript.

2. Lines 24: The study lacks a control group for definitive clinical conclusions. Please justify the lack of a control group and conclusions of the study.

Thank you for your insightful suggestion about the lack of a control group in our study. While we acknowledge that the inclusion of a control group would strengthen the clinical conclusions, this study was conducted as a retrospective analysis of real-world clinical data, focusing on patients who underwent combined IPL and MGX therapy as part of their routine care. We will respectfully incorporate this point as a limitation in the discussion section of our manuscript and emphasize the importance of conducting future randomized controlled trials to validate these results.

3. Lines 106: Please clarify this “Lipid-layer image analysis”, aspect of the study. It’s not clear whether the authors are assessing the colors changes or deriving the thickness of TFLL from the interferometric patters obtained by the Keratograph 5M. While its clear that color changes equate to LLT please explain for readers how the changes equate to thickenss.

Thank you for your valuable feedback. We have additionally detailed the formula by employing the previously described method of analyzing image colors and using the Euclidean formula to identify the closest matching similarity.

4. Furthermore, is the Total LLT= LLT inferotemporal+LLT superotemporal + LLT inferonasal + LLT superonasal or the mean of all these regions?

The total LLT refers to the LLT of the entire area before image segmentation. We will clarify this further in the manuscript.

5. Please justify the use of lower half of the cornea for image analysis and its clincal significance.

To ensure accuracy in the examination, the lower cornea was measured, as the upper cornea is often influenced by the eyelids or eyelashes, leading to a higher likelihood of inaccurate measurements. We will specify this in the manuscript.

6. It would be more substantial if the authors could compare their LLT with devices such as Lipiview, etc. If not possible please discuss.

As this was a retrospective study, it was not possible to compare the results with LipiView. We will include this as a limitation in the manuscript.

7. Table 2 shows the changes in various clinical characteristics of dry eye, it hasn’t been mentioned how these changes effects the LLT regions or Horizontal difference of LLT. Possible this might bring out the clinical significance of this study, any correlation or risk factors. Please include it in the discussion:

a. Song, Yilin, et al. "Tear film interferometry assessment after intense pulsed light in dry eye disease: A randomized, single masked, sham-controlled study." Contact Lens and Anterior Eye 45.4 (2022): 101499.

b. Ahmed, S. et al. (2019). Effect of intense pulsed light therapy on tear proteins and lipids in meibomian gland dysfunction. Journal of ophthalmic & vision research, 14(1), 3.

c. Zhao, Hui, et al. "Lipidomics profiles revealed alterations in patients with meibomian gland dysfunction after exposure to intense pulsed light." Frontiers in Neurology 13 (2022): 827544.

We appreciate your insightful comments regarding the correlation between changes in dry eye parameters and the lipid-layer thickness (LLT) regions or horizontal differences of LLT. While our study primarily focused on the vertical LLT differences and overall distribution, we acknowledge the importance of exploring potential correlations between the clinical characteristics of dry eye disease (DED) and regional LLT changes.

In light of this, we will incorporate a discussion on the possible clinical implications of these findings, referencing the studies you have suggested (Song et al., 2022; Ahmed et al., 2019; Zhao et al., 2022). Specifically, we will analyze the relationship between significant changes in clinical parameters, such as the ocular surface disease index (OSDI) and meibomian gland expressibility, with the vertical and horizontal LLT differences observed in our results. While our study did not specifically examine risk factors or direct correlations, future studies could expand on this aspect by including additional analyses of tear film lipidomics and detailed protein profiles, as highlighted by Ahmed et al. and Zhao et al.

We will also highlight the clinical significance of our findings by discussing how the improved LLT uniformity, particularly in the upper and lower regions, might reflect enhanced tear film stability and reduced lipid-layer stasis. This aligns with Song et al.'s emphasis on IPL-induced changes in lipid-layer dynamics and their contribution to clinical

Thank you for your constructive feedback, which has allowed us to strengthen the discussion and provide a more comprehensive understanding of the clinical significance of our find

8. Figure 1 should a detailed “study design and process” figure.

Thank you for the great suggestion. We have added Figure 1, which illustrates the study protocol.

9. While figure 1 and figure 2 are interesting, these are averaging the pixels, please state clearly how many pixels were used to make each quadrant. How did the authors eliminate biases while performing this analysis as not all eyes are of the same size and the focal length of the patient is not regular. Furthermore, the iris of patients are of various color intensities.

Thank you very much for your excellent feedback. The image size was controlled following the guidelines of previous studies. All images were cropped uniformly using the 9th Placido ring as a reference for analysis. For quadrant analysis, the images were evenly divided into upper, lower, left, and right sections using Python. To avoid any confusion, we will add more detailed descriptions of these methods in the Methods section of the manuscript. Thank you once again for your valuable input.

10. Figure 2 contains 3 panels, please simplify by labeling them as A, B, C. Same with figure 1 that has 2 panels.

Yes, the panels were divided into A, B, and C. Thank you.

11. Figure 2 C is a line graph that has to has to have more description. i.e the x-axis and y-axis are not apparently clear to the reader.

Thank you. Explanations for the x-axis and y-axis have been added to the graph.

12. Lines: 204. The hypothesis has not taken into consideration that the upper half of the cornea was not assessed and what are the pro and cons of this approach.

 Thank you for your valuable feedback. To fully validate the hypothesis, the lipid layer of the upper cornea would also need to be analyzed. We will include this as a limitation in the discussion section.

13. As stated in lines 224 this is a novel study, however, the lipid layer is a structure floating on the aqueous and mucin layers, the author has to assess the interplay of the various layers and whether the other layers (TMH volume, TBUT integrity, MG expressibility, MGquality had any influence on their findings.

Thank you for your valuable comment. While our study focused on changes in the lipid layer, we also evaluated related parameters such as TMH, TBUT, and meibomian gland expressibility and quality, as shown in Table 2. The significant improvement in expressibility and quality suggests enhanced meibomian gland function, which likely contributed to the lipid layer changes. We acknowledge the importance of assessing the interplay between all tear film layers and will include this as a limitation and a direction for future research in the discussion section.

Reviewer #2: I sincerely appreciate the opportunity to read and evaluate the authors research on tear lipid layer with IPL and meibomian gland expression. I’ve been privileged to review several dozen dry eye papers to date in addition to publishing several on the topic; in light of this, and although there may be hetereogeneity in measurements, patients, and follow up, this paper ranks among the top 1% on the topic in terms of scientific rigor and methodology. Very impressive, and a congratulations to the authors are in order.

We are deeply grateful for your kind words about our study and truly appreciate your valuable review.

Please see below for my comments:

Abstract: Line 50 should read“is a widely used treatment…” remove the word “most”

 Thank you. We removed “most”

Intro does a great job at laying out the landscape of dry eye and adequately addresses the niche and need for this paper.

Methodology includes IRB and ethics review.

Dry eye protocols are quite rigorous and standardized, very impressive.

Language is concise.

 Thank you for your valuable feedback.

Results: While it does say who did the IPL, does it say who did the exams?

 The slit-lamp examination and IPL were performed by a single experienced specialist, and the Placido tear film analyzer was operated by a single skilled technician. We will add this information to the manuscript. Thank you.

The sole critique and question that I have is maybe the authors should include the NNT statistic based on the primary outcome. Also for those statistically significant differences on secondary outcomes was Bonferroni performed? If not, you may see that statistical significance disappeares due to the small difference between the groups, and the fact that it helps to adjust p-values and mitigate the risk of Type I errors arising from conducting numerous statistical tests on the same dataset.

Otherwise a brilliant study and look forward to the response.

Thank you for your excellent advice. We have added the sample power calculation details to the statistics section and indicated in the result figure that the p-values were adjusted based on the Bonferroni correction. We truly appreciate your thoughtful evaluation of our study.

Reviewer #3: Friday, January 3

Dear Editor

Thank you for the invitation to review the manuscript titled: Changes in the distribution of the tear lipid layer after intensive pulse light combined meibomian gland expression in patients with meibomian gland dysfunction.

The aim of the study is to investigate changes in dry eye disease (DED) parameters and lipid-layer distribution after treatment with intensive pulse light combined with Meibomian gland expression treatment in patients with Meibomian gland dysfunction (MGD). After reviewing the manuscript, I think it requires a major revision to be suitable for publication in your admired journal. However, some comments to the author are presented below to improve the manuscript:

Thank you for your valuable review and excellent feedback. We will faithfully implement your suggestions to enhance the quality of the study.

Materials and Methods

1. Page 5, lines 39-35: please clear the treatment periods (What is the interval between sessions, and why were the measurements done two weeks after IPL, not earlier?

Thank you for your valuable feedback. I will ensure to specify the duration in the IPL protocol section. This retrospective study generally follows the approach I use in clinical practice. Overall, the intervals are based on the study by Dr. Toyos (PLoS One. 2022 Jun 23;17(6):e0270268), which implemented a protocol of four sessions at two-week intervals, followed by an examination four weeks later. However, in my practice, I typically schedule three sessions at two-week intervals, taking into account patient compliance.

Furthermore, to avoid potential distortion in examination results caused by excessive meibum expression immediately after IPL + MGX, I perform the examination during the fourth visit, two weeks after the third session, when the results have stabilized. Additional treatments are provided as needed based on the patient's condition. I will incorporate this information more thoroughly into the manuscript.

2. The method should explain the light parameters used during IPL therapy using an acne filter, such as the emitted light wavelength, light intensity, operating voltage, power, frequency, maximum optical energy, pulse duration, and repetition time or any data regarding the instrument used.

The IPL protocol we used in this study is primarily based on Dr. Toyos' protocol and has been adjusted according to updates from the manufacturer and recent studies. We used acne filter emitting wavelengths of 400–600 nm and 800–1200 nm to minimize the impact on melanocytes. The intensity is initially set at 10~12 J/cm², guided by Fitzpatrick skin typing, and then adjusted as needed. We described additional details regarding power, pulse count, filter, and intensity in the methods section. Thank you.

3. The author mentioned in the abstract that: This retrospective study included 218 patients diagnosed with MGD, while in the result section (Page 7, line 128), he mentioned the number of patients was 337. In addition, the number of patients should be added obviously in the method section.

Out of the 337 patients who underwent the examination, 218 met the inclusion criteria. We have added the number of patients to the methods section and revised the wording to avoid any misunderstanding in the results. Thank you.

4. In the discussion section, pages 10-11, from lines 182-194, there are obvious overlaps. These paragraphs can be summarized in one or two sentences only.

Thank you for your feedback. We have removed redundant content and condensed it into two sentences. Thank you again.

5. Page 13, lines 232-236: there was no relation between the two sentences about the treated eyes and the healthy individuals. Please try to find another reference to support your data.

Thank you for your precise feedback. The two sentences were mistakenly placed in separate paragraphs, and this has been corrected. Previous studies on differences in conjunctival hyperemia between the nasal and temporal regions have only been conducted in healthy individuals. As per your suggestion, I will state that this study raises the possibility of such differences in dry eye patients and highlight the need for further research. I sincerely appreciate your thoughtful review.

6. Page 13, lines 241-242. T

---

## [Decision Letter · Decision Letter 1]

5 Feb 2025

Changes in the distribution of the tear film lipid layer after intensive pulsed light combined with meibomian gland expression in patients with meibomian gland dysfunction

PONE-D-24-24404R1

Dear Dr. Choi,

We’re pleased to inform you that your manuscript has been judged scientifically suitable for publication and will be formally accepted for publication once it meets all outstanding technical requirements.

Kind regards,

Kofi Asiedu, O.D., Ph.D.

Academic Editor

PLOS ONE

Additional Editor Comments (optional):

Reviewers' comments:

Reviewer's Responses to Questions

**Comments to the Author**

1. If the authors have adequately addressed your comments raised in a previous round of review and you feel that this manuscript is now acceptable for publication, you may indicate that here to bypass the “Comments to the Author” section, enter your conflict of interest statement in the “Confidential to Editor” section, and submit your "Accept" recommendation.

Reviewer #1: All comments have been addressed

Reviewer #2: All comments have been addressed

Reviewer #3: All comments have been addressed

2. Is the manuscript technically sound, and do the data support the conclusions?

Reviewer #1: Yes

Reviewer #2: Yes

Reviewer #3: Yes

3. Has the statistical analysis been performed appropriately and rigorously? 

Reviewer #1: Yes

Reviewer #2: Yes

Reviewer #3: I Don't Know

4. Have the authors made all data underlying the findings in their manuscript fully available?

Reviewer #1: Yes

Reviewer #2: Yes

Reviewer #3: Yes

5. Is the manuscript presented in an intelligible fashion and written in standard English?

Reviewer #1: Yes

Reviewer #2: Yes

Reviewer #3: Yes

6. Review Comments to the Author

Reviewer #1: I have no comments; the authors seems to have addressed all previous concerns to the best of their ability.

Reviewer #2: Authors have meticulously addressed all my concerns, and by the looks of it, the other editors' concerns as well.

Reviewer #3: Monday, January 27

Dear Editor

It is an honor to have been invited by you to re-review the manuscript “Changes in the distribution of the tear lipid layer after intensive pulse light combined meibomian gland expression in patients with meibomian gland dysfunction.”

I appreciate the opportunity to contribute to improving this manuscript as the author followed the comments point by point. After these revisions, the manuscript is suitable to be accepted in your admired journal “Plos One.”

Best regards

7. PLOS authors have the option to publish the peer review history of their article (what does this mean? ). If published, this will include your full peer review and any attached files.

**Do you want your identity to be public for this peer review?** For information about this choice, including consent withdrawal, please see our Privacy Policy .

Reviewer #1: No

Reviewer #2: No

Reviewer #3: No

---

## [Editor Report · Acceptance letter]

PONE-D-24-24404R1

PLOS ONE

Dear Dr. Choi,

I'm pleased to inform you that your manuscript has been deemed suitable for publication in PLOS ONE. Congratulations! Your manuscript is now being handed over to our production team.

Kind regards,

on behalf of

Dr. Kofi Asiedu

Academic Editor

PLOS ONE